# Relationships between Body Weight Status and Serum Levels of Adipokine, Myokine and Bone Metabolism Parameters in Healthy Normal Weight and Thin Children

**DOI:** 10.3390/jcm11144013

**Published:** 2022-07-11

**Authors:** Jadwiga Ambroszkiewicz, Magdalena Chełchowska, Joanna Mazur, Grażyna Rowicka, Joanna Gajewska

**Affiliations:** 1Department of Screening Tests and Metabolic Diagnostics, Institute of Mother and Child, 01-211 Warsaw, Poland; magdalena.chelchowska@imid.med.pl (M.C.); joanna.gajewska@imid.med.pl (J.G.); 2Department of Humanization in Medicine and Sexology, Collegium Medicum, University of Zielona Gora, 65-729 Zielona Gora, Poland; joanna.mazur@hbsc.org; 3Department of Nutrition, Institute of Mother and Child, 01-211 Warsaw, Poland; grazyna.rowicka@imid.med.pl

**Keywords:** adipokines, myokines, bone metabolism markers, bone mineral density, prepubertal children

## Abstract

Optimal body weight and body composition for age are relevant to child development and healthy life. Changes in lean mass and fat mass as well as its distribution are associated with alterations in the secretion of myokines and adipokines by muscle and adipose tissues. These factors are very important for bone health. The aim of the study was to assess serum leptin, adiponectin, resistin, visfatin and omentin as adipokines and myostatin and irisin as myokines with regard to their associations with bone parameters in healthy normal weight and thin children. We studied 81 healthy prepubertal children (aged 5 to 9 years) divided into three groups: group A—35 children with a BMI z-score between +1 and −1 SD; group B—36 children with a BMI z-score between −1 and −2 SD; and group C—10 thin children with a BMI z-score of <−2 SD. We observed significantly (*p* < 0.001) lower fat mass, fat/lean mass ratio and bone mineral density (BMD) across weight status with the lowest values in the group of thin children. We noticed significantly (*p* < 0.05) lower concentrations of 25-hydroxyvitamin D, resistin and high-molecular-weight (HMW) adiponectin but higher levels of myostatin as the BMI z-score deceased. We found that BMI and leptin levels were directly correlated with fat mass, lean mass, bone mineral content (BMC) and BMD. Resistin levels were negatively associated with lean mass, while visfatin concentrations were positively related to total BMD. In healthy prepubertal children there were differences in body composition and in bone mineral density across decreasing BMI status. We suggest that changes in serum myostatin and 25-hydroxyvitamin D levels may play a role in bone status of thin children. Moreover, significant relations between adipokines and bone parameters may confirm crosstalk between fat tissue and bone in these children.

## 1. Introduction

Skeletal muscle and bone are tightly connected anatomically and physiologically and play a crucial role in locomotion and metabolism. Additionally, adipose tissue is vital in this biochemical crosstalk [1,2,3]. These three organs share a common mesenchymal precursor and communicate through the endocrine system, secreting a family of cytokines, namely myokines (from muscle), adipokines (from adipocytes) and bone markers (from bone cells) [4,5].

Among several myokines synthesized by skeletal muscle, myostatin and irisin appear to be well known and have promising clinical value. Myostatin belongs to the of transforming β growth factor (TGF-β) family and plays a role in the regulation of skeletal muscle mass and bone formation [6]. It suppresses muscle growth and differentiation by binding to the activin receptor on the muscle membrane [7,8]. Myostatin also accelerates osteoclastogenesis mediated by the receptor activator for nuclear factor ĸB ligand (RANKL). Irisin is a peptide produced in response to physical activity cleaved from the extracellular domain of fibronectin type III domain containing proteins 5 (Fndc5) by stimulation with peroxisome proliferator-activated receptor gamma 1-α (PGC-1α), which is expressed in the skeletal muscle and other tissues [9,10]. Irisin benefits on skeletal muscles by promoting myogenic differentiation and myoblast fusion. Despite the osteogenic effect of irisin, no correlation has been found between serum concentration of this protein and the markers of bone turnover in the study of adults [11].

Adipose tissue can secrete a series of cytokines named adipokines. Some of them, such as leptin, resistin and visfatin, have proinflammatory properties. Leptin, the best characterized adipokine, which acts directly through its receptors, plays a role in regulating appetite and energy consumption and in bone metabolism through its paracrine or autocrine action. Clinical studies have shown that serum leptin concentrations were proportional to total body weight and fat mass [12]. Resistin is mainly secreted by macrophages and monocytes but also by adipose tissue and skeletal muscle [13]. It plays an essential role in inflammatory responses and has been linked to several chronic diseases and obesity. Visfatin, expressed mainly by visceral adipose tissue, regulates adipocyte differentiation and is involved in the control of body weight [14,15]. Serum visfatin concentrations depend on lifestyle interventions, including exercise and diet. As a multifunctional protein, visfatin may act as a hormone, cytokine, and enzyme-nicotinamide phosphoribosyltransferase (Nampt) [16].

Fat tissue releases also anti-inflammatory adipokines, including adiponectin and omentin. Adiponectin, which structurally belongs to the collagen superfamily protein, is involved in many physiological processes, including lipid and glucose metabolism, insulin resistance, inflammation and bone metabolism. Studies have shown that serum levels of adiponectin inversely correlate with body mass index (BMI), fat mass and bone mineral density in children and adults [17]. In serum, adiponectin circulates in isoforms. Among them, high-molecular-weight (HMW) adiponectin seems to be an active form and the ratio of HMW/total adiponectin is a better indicator of metabolic disturbances than total adiponectin [18]. Omentin is an anti-inflammatory and anti-atherosclerotic glycoprotein secreted from visceral adipose tissue. It plays a role in adipocyte differentiation and maturation, insulin resistance and immune response regulation [19].

Bone growth during childhood is critical for attaining peak bone mass at skeletal maturation in early adulthood. Markers of bone metabolism have been associated with muscle mass and function. It is known that vitamin D, which regulates calcemic status, has pleiotropic effects on bone cell differentiation, promoting either bone formation or bone resorption. Vitamin D also affects muscle development by mediating gene expression of myogenic transcription factors and contractile proteins, regulating membrane calcium channel and mitochondrial oxidative functions [20]. Biochemical parameters helpful in assessing the rate of bone turnover include osteocalcin (OC) as a marker of bone formation and cross-linked C-terminal telopeptide of type I collagen (CTX) as a marker of bone resorption [21]. 

It is important to note that optimal body weight and body composition for age, including lean mass, fat mass and bone mass, are relevant to child development and healthy life. Low body mass may promote muscle wasting and also affect bone mass. Changes in body composition, especially in lean mass and fat mass as well as its distribution are associated with alterations in the secretion of myokines and adipokines by muscle and adipose tissues. These factors influence bone metabolism, leading to decreased bone mass. 

So far, there have been limited reports regarding body composition and myokine and adipokine status in underweight children. It is an important problem because the prevalence of thinness among children has increased according to data from several European countries [22,23].

The aim of this study was to investigate the associations of circulating myokines, adipokines and bone turnover markers with body composition and bone mineral density in healthy normal weight and thin children. 

## 2. Materials and Methods

### 2.1. Subjects

The studied children were recruited between April 2017 and May 2018 from a group of consecutive patients at the Department of Nutrition at the Institute of Mother and Child in Warsaw (Poland). These children were apparently healthy, without endocrine disorders or genetic syndromes, without diseases that could affect nutritional status, growth and pubertal development, without a history of fractures, and not receiving medications except for routine vitamin D supplementation (at a median dose of 1000 IU/day (range 600–1500 IU/day). All children remained on a traditional omnivorous diet. Pubertal stage was assessed according to the Tanner scale. Children classified as Tanner stage 1 were included in this study. The calculated weight for age BMI z-score was based on the World Health Organization’s (WHO) standards [24,25] and assessed using macro SPSS AnthroPlus.

In total, we examined 81 Caucasian prepubertal children aged 5–9 years divided into three groups: group A—children with a BMI z-score between +1 and −1 SD (*n* = 35); group B—with a BMI z-score between −1 and −2 SD (*n* = 36); and group C—thin children with a BMI z-score of <−2 SD (*n* = 10). Inclusion criteria were: BMI z-score between +1 and <−2 SD (prepubertal period). Overweight or obese children as well those who showed pubertal development were excluded from this study.

The studied children were attaining the WHO recommendation regarding physical activity (WHO guidelines on physical activity and sedentary behavior. Geneva. WHO 2020). According to obtained questionnaire, they accumulated about 60–90 min/day of MVPA (moderate-to-vigorous physical activity) and approximately 30 min/day of VPA (vigorous physical activity). VPA included activities after school twice a week for 1 or 2 h. There were no significant differences in the levels of physical activity in examined groups of children.

The study was conducted in accordance with the Declaration of Helsinki and approved by the Ethics Committee of the Institute of Mother and Child (protocol code: 12/2017, date of approval 30 March 2017). Written informed consent has been obtained from all parents of the children to publish this paper. 

### 2.2. Anthropometric Measurements

Measurements of body height and weight were performed in all participants using standard procedures. Body height was measured with a stadiometer and recorded with a precision of 0.1 cm, and weight was assessed unclothed with a calibrated balance scale to the nearest 0.1 kg. Body mass index was calculated as body weight (kg) divided by height squared (m2). Fat mass (in percentage and in grams), lean mass, bone mineral content (BMC) and bone mineral density (BMD) in the total body and in the lumbar spine (L2-L4) were measured by dual-energy X-ray absorptiometry (DXA) using Lunar Prodigy (General Electric Healthcare, Madison, WI, USA). We obtained BMD z-scores using the pediatric reference population database encore software version 9.30.044, which allows the use of nationality-based reference data. The subjects’ BMDs were compared with the average BMD of a reference group of the same age, sex and ethnicity. These parameters were measured by trained staff with the same equipment, using standard positioning techniques.

### 2.3. Blood Analyses

For biochemical measurements, venous blood samples were collected in the morning hours between 8:00 and 10:00 AM after an overnight fast to avoid diurnal variations. The samples were centrifuged at 1000× *g* for 10 min at 4 °C and obtained serum (in portions) was stored in −80 °C until assay. Biochemical parameters were assessed in all studied children, except of omentin concentration, which was determined in 69 (85%) subjects. Serum concentration of 25-hydroxyvitamin D was determined by electrochemiluminescent immunoassay (ECLIA) using kits from DiaSorin Inc. (Stillwater, OK, USA) on a Liaison analyzer with a precision coefficient of variance (CV): 6.0–9.8%. Concentrations of myokines (myostatin, irisin), adipokines (leptin, resistin, visfatin, adiponectin, omentin) and bone turnover markers (OC, CTX) were assessed using enzyme-linked immunosorbent assay (ELISA), according to the manufacturer‘s instructions. 

Serum myostatin and omentin concentrations were determined using kits from SunRed Biotechnology (Shanghai, China) with intra- and interassay coefficients of variation (CVs) less than 8% and 11% for myostatin and less than 10% and 12% for omentin, respectively. Irisin levels were assayed using a kit from BioVendor (Brno, Czech Republic). The intra- and interassay CVs in this method were 4.9–8.2% and 8.0–10.7%, respectively. Leptin and resistin levels were assessed using commercial kits from DRG Diagnostics (Marburg, Germany), with intra-assay CV of 5.95% and inter-assay CV of 8.66% for leptin and intra-assay CV of 5.2–6.6%, inter-assay CV of 7.0–8.1% for resistin. Serum visfatin levels were determined using Nampt (Visfatin/PBEF) (human) ELISA kit from Adipogen Life Science (Liestal, Switzerland), with intra-assay CV of 2.31–9.11% and inter-assay CV of 4.66–7.24%. Adiponectin concentrations were detected using HMW and Total Adiponectin ELISA kit from ALPCO (Salem, USA), where intra-assay CV was 5.3–5.4% for total adiponectin and 3.3–5.0% for HMW adiponectin, and inter-assay CV was 5.0% for total adiponectin and 5.7% for HMW adiponectin. 

Concentrations of OC and CTX were detected using N-MID Osteocalcin and Serum CrossLaps (CTX-I) kits from Immunodiagnostic Systems (Boldon, UK). The intra- and interassay CVs were: 1.3–2.2% and 2.7–5.1% for OC and 1.7–3.0% and 2.6–10.9% for CTX, respectively. 

### 2.4. Statistical Analyses

The obtained data were statistically analyzed using IBM Statistics for Windows version 27.0 software (Amonk, NY, IBM Corp.). All variables were tested for normality using the Kolmogorov–Smirnov test. Data are presented as frequency (percentage), means ± standard deviation (SD) for normally distributed data or medians and interquartile ranges 25th–75th (IQR) for skewed distribution. Group differences were assessed using the non-parametric Kruskal–Wallis H test or the Mann–Whitney U test, as appropriate. A univariate correlation analysis was performed using the Spearman (rho) coefficient. A *p*-value of less than 0.05 was considered statistically significant.

## 3. Results

The data on the children’s characteristics and anthropometry are presented in Table 1. We examined three groups of children: group A (*n* = 35, 17/48.6% F, 18/51.4% M), BMI z-score between +1 and −1 SD; group B (*n* = 36, 17/47.2%F, 19/52.8% M), BMI z-score between −1 and −2 SD; and group C (*n* = 10, 4/40% F, 6/60% M), BMI z-score < −2 SD. The studied groups of children were similar in terms of age, sex distribution and height. We observed significantly lower fat mass and the ratio of fat to lean mass across weight status with those values being the lowest in the group of thin children. Additionally, the values for BMD in the total body and in the lumbar spine L2-L4 as well as BMD z-score BMD L2-L4 z-score were significantly decreased in groups B and C compared with children from group A (*p* < 0.001; *p* < 0.001, *p* < 0.01, *p* < 0.01, respectively). Total BMC was also decreased between the studied groups; however, these differences did not reach statistical significance. 

Regarding the biochemical parameters, we found significantly lower concentrations of resistin (*p* < 0.05), HMW adiponectin (*p* < 0.05) and the ratio of HMW to total adiponectin (*p* < 0.001) in the groups as the BMI z-score deceased (Table 2). Although median values of serum omentin, total adiponectin and visfatin were lower by about 15–35% in thin children, these differences were not statistically significant. Among myokines, we noted increased concentrations of serum myostatin (*p* < 0.05) with lowering BMI z-scores, while irisin levels were comparable. We did not observe significant differences in serum concentrations of bone turnover markers (osteocalcin and CTX) in the studied groups, except for lower 25-hydroxyvitamin D levels in groups B and C compared with the children from group A (*p* < 0.05). 

Additionally, we assessed the ratios of anti-inflammatory to pro-inflammatory adipokines in two groups (A and B) which were comparable in terms of the number of studied children (Table 3). In the thinner children, we found significantly (*p* < 0.05) lower ratio of adiponectin to resistin and a higher ratio of omentin to leptin. However, there were no statistically significant differences in the ratios of adiponectin/leptin and omentin/resistin between the examined groups. 

As expected, we found significant positive correlations between BMI and fat mass, lean mass, BMC and BMD in the whole group of children (Table 4). In addition, fat and lean mass positively correlated with BMC, total BMD and BMD L2-L4. Among adipokines, leptin levels were directly correlated with fat mass, lean mass, BMC and BMD, and resistin levels were negatively associated with lean mass. We did not observe significant associations between serum levels of bone metabolism markers and body composition or bone mineral density parameters, except for positive correlations (*p* < 0.05) between 25-hydroxyvitamin D levels and fat mass and between OC levels and fat mass. Serum concentrations of myokines did not correlate with body composition or bone mineral density. 

Additionally, in the whole group of children, we observed positive correlations between BMI and 25-hydroxyvitamin D (r = 0.317, *p* = 0.004), leptin (r = 0.266, 0.017) and HMW adiponectin levels (r = 0.263, *p* = 0.018). We also found significant weak correlations between biochemical parameters: omentin and irisin levels (r = 0.290, *p* = 0.044), omentin and 25-hydroxyvitamin D (r = −0.315, *p* = 0.023) and resistin and OC (r = −0.237, *p* = 0.037) (data not shown in the tables).

We also analyzed correlations between the ratios of anti- to pro-inflammatory adipokines and anthropometric parameters, bone markers and myokines in the whole group of studied children (Table 5). The adiponectin to leptin (A/L) ratio was negatively correlated with BMI, lean mass, BMC and total and lumbar spine BMD. The omentin to leptin (O/L) ratio was inversely related to total BMD, while the omentin to resistin (O/R) ratio was negatively correlated with fat mass, fat/lean ratio and serum 25-hydroxyvitamin D levels and positively correlated with levels of myostatin and irisin.

## 4. Discussion

Optimal body weight as well as proper body composition in childhood are very important for bone health [26,27]. This is evidenced by highly significant direct correlations between body mass index, fat mass and lean mass and their associations with bone mineral content and bone mineral density. In the present study, we observed significantly lower body composition parameters such as fat mass and the fat to lean mass ratio as well as the values of BMD in the total body and in the lumbar spine L2-L4 across weight status, with the lowest values in the group of thin children. Studies on the relationship between weight and bone mineralization focus mainly on patients with obesity [28]. Underweight as well obese subjects have disturbances in bone metabolism. Children with obesity have a significantly greater bone mass (BMC, BMD) compared with normal weight subjects [29]. Sanchez Ferrer et al. [30] observed lower BMC and BMD in underweight children than in normal weight or overweight ones. Subjects with low body mass generally have decreased bone mineral density parameters. About 44% of adult subjects with constitutional thinness and about 50% patients with anorexia nervosa presented osteopenia [31,32]. 

Measurements of bone mineral density using densitometry techniques are recognized as the gold standard of assessing bone status. However, assessment of biochemical bone turnover marker levels, which reflect the dynamics of bone formation and resorption processes, is also valuable. In the present study, we did not find differences in the concentrations of bone turnover markers such as CTX and OC. We did not observe relationships between CTX levels and bone density parameters or with adipokine or myokine levels in prepubertal children. Kirk et al. [21] observed that higher CTX levels were associated with bone loss and poorer muscle function in older people but showed poor diagnostic power for these measures. 

Osteocalcin synthetized in osteoblasts is one of the bone formation markers involved in bone matrix mineralization. Carboxylated osteocalcin (cOC) is an active form of osteocalcin in bone [33]. In our previous study, although we did not find differences in serum levels of total OC between thin and normal weight children, we noticed comparable cOC concentrations and significantly higher levels of the undercarboxylated form of osteocalcin (ucOC) [34]. This leads to a lower ratio of c-OC to uc-OC in thin children. Additionally, other authors found higher ratios of uc-OC/c-OC in leaner than in heavier children, suggesting a weight-dependent association between forms of osteocalcin [35]. Osteocalcin also improves muscle function and may influence fat mass in terms of increased anti-inflammatory cytokines and decreased pro-inflammatory cytokines [36]. In skeletal muscle, osteocalcin promotes the uptake and catabolism of glucose and fatty acids and is also responsible for an increase in IL-6 levels, a myokine that promotes adaptation to exercise [37]. In our study, osteocalcin level was significantly correlated both with fat mass and lean mass and with resistin as well as the adiponectin to resistin ratio. Our results and data from other researcher may suggest that osteocalcin affects not only bone but also muscle and fat tissue metabolism. 

Vitamin D has direct and indirect (through Ca and P) effects not only on bone but also on skeletal muscle and its deficiency-accelerated muscle atrophy [38,39]. Vitamin D status is associated with under nutrition and low serum 25-hydroxyvitamin D levels were more common in underweight children than normal weight [40]. We also observed significantly lower 25-hydroxyvitamin D status in leaner children compared with heavier children. In addition, we noticed that vitamin D was positively correlated with BMI and fat mass and inversely with omentin levels as well as with the omentin/resistin ratio in prepubertal children. Vitamin D deficiency is also more common in obese subjects and can be caused by many factors, such as reduced exposure to sunlight due to reduce physical activity or increased deposition of vitamin D in fat tissue [41]. According to Gilbert-Diamond et al. [42] serum level of 25-hydroxyvitamin D was inversely correlated with the amount of adipose tissue in children. The unfavorable status of vitamin D in patients with obesity may also be related to 1.25 dihydroxyvitamin D deficit in association with high leptin levels [43]. It is suggested that leptin, through fibroblastic growth factor (FGF-23), inhibits the synthesis of the active form of vitamin D, reducing the expression of cholecalcifediol 1α-hydroxylase, which catalyzes the hydroxylation of vitamin D in the kidneys. The mechanism of vitamin D deficit seems to be different in thin, normal weight and obese children. 

Skeletal muscle secretes many active molecules named myokines, which not only play a role in muscle metabolism but also influence other organs such as bone and adipose tissue. It is important to note that physical activity (PA) is the primary physiological stimulus for bone metabolism through the production of myokines. In addition, the exercise-induced myokines can exert an anti-inflammatory action that is able to counteract not only acute inflammation (due to an infection) but also the chronic, low-grade inflammation underlying many chronic diseases [44,45]. In our study, there were no differences in either MVPA or VPA between the studied groups. There have been limited studies regarding myokine levels in prepubertal children. Only Elizondo-Montemayor et al. [46] assessed irisin levels and observed its lower concentrations in underweight children compared with normal weight and obese groups. In our study, we did not observe significant differences in the serum concentration of irisin in the context of BMI status. Although we did not find significant correlations between irisin and anthropometric parameters, we observed that irisin levels positively correlated with omentin concentrations. It is worth noting that irisin acts in a pleiotropic manner and is not only a myokine but also an adipokine secreted by fat tissue [47].

In the present study, we noticed steadily increased serum levels of myostatin in relation to different BMI, with the highest in thin children. Myokines did not correlate with anthropometric parameters, but in the small group of thin children, we noticed an inverse correlation between myostatin and 25-hydroxyvitamin D levels (r = −0.723, *p* = 0.018) (data not shown in the tables). In thinner children we also observed low vitamin D status. 

There is limited knowledge on the adipokine profile in prepubertal children with different body weight status [16,17]. Researchers showed high leptin and low adiponectin levels in overweight and obese children, but results regarding other adipokines are controversial [29]. In underweight patients with anorexia nervosa Seidel et al. [16], did not show significant differences in visfatin levels compared with healthy controls. Vehapoglu et al. [17] found significantly lower vaspin and apelin levels in underweight children compared with controls; however, visfatin levels were similar. Redondo et al. [48] reported significantly lower serum leptin, visfatin and chemerin levels and higher total adiponectin and omentin levels in lean children compared with obese children. As leptin, visfatin and chemerin are proinflammatory adipokines, while adiponectin and omentin have anti-inflammatory properties, the authors observed decreased proinflammatory profiles of circulating adipokines in lean children.

We did not find significant differences in leptin, visfatin, adiponectin and omentin levels between the studied groups. However, we observed a gradual decrease in resistin concentrations in children with lower BMI and its negative correlation with lean mass. Contrary to leptin, adiponectin is associated with lower total body mass and regional bone mass in adulthood. An inverse association between adiponectin and bone mass in childhood has also been found [49]. Sayers et al. [13] suggested that setting of the adiponectin axis in childhood may relate to subsequent bone development, skeletal architecture, bone strength and fracture risk. Although total adiponectin did not significantly differ between our studied groups, its active isoform, HMW adiponectin, as well as the ratio of HMW to total adiponectin, were the lowest in thin children. A partial explanation could be related to the disruption of adiponectin isomerization in thin children, which may influence not only its proper action but also bone mass.

Assessing the anti-inflammatory and pro-inflammatory properties of adipokines, we found significantly lower ratio of adiponectin to resistin and a higher ratio of omentin to leptin in the group of thinner children. In the correlation analysis, we observed significantly negative correlations of the A/L ratio with BMC and BMD (in the total body and in the lumbar spine). The O/L ratio was inversely associated with total BMD. Additionally, Tamme et al. [50] reported that the leptin to adiponectin ratio is negatively associated with lumbar spine BMD in puberty. This finding indicates that both levels of adipokines synthetized by fat tissue and also the proportion of pro- and anti-inflammatory adipokines are important in the context of the modulatory effects on both the inflammatory process and bone metabolism.

Early recognition of underweight with the accompanying consequences on muscle, bone and adipose tissue metabolism are crucial for the achievement of optimal bone mass. Monitoring of anthropometric and biochemical parameters, including bone markers, myokine and adipokine profiles, may be helpful in pediatric practice to prevent bone loss and muscle deficits. Reference values of myokines, adipokines and bone turnover markers in the pediatric population should be established to improve their clinical application. We believe that our study could add new evidence to the understanding of the muscle–bone–fat axis; however, future studies are necessary to fully understand these associations.

The presented study has several limitations. First, our results come from a cross-sectional study that does not permit causality statements. However, it is the first study assessing myokines, adipokines and bone parameters in groups of prepubertal children with different BMI z-scores. Second, our study was limited to prepubertal children who had data (densitometry and biochemical parameters) measured at the same time. Thus, our sample size was relatively small, especially for the group of thin children. However, the studied group of children was homogenous in terms of ethnicity (all children were Caucasian) and development period and was clinically well characterized. Large groups of healthy children are rarely studied because of ethical considerations regarding blood sampling in healthy children. Third, we used BMI z-scores to differentiate the groups of children. Although not ideal, the use of BMI is the most practical method currently available due to being based on common anthropometric measures of weight and height used in many studies. Fourth, we did not obtain detailed data regarding diets and physical activity (only a questionnaire) for the studied subjects. However, these children were generally healthy, without eating disorders, on an omnivorous diet and with comparable levels of physical activity. Fifth, our results were based on only single measurements of biochemical markers and therefore may not reflect their long-term exposure. In the total sample, 15% of the studied subjects had missing data for omentin concentrations and that is why we did not assess the ratio of pro-inflammatory to anti-inflammatory adipokines in the small group of thin children (Table 3). Among myokines, we determined only myostatin and irisin levels; however, we are planning to assess novel myokines such as myonectin, decorin and FGF-21 in our future study.

In conclusion, our results demonstrate that in healthy prepubertal children there were significant differences in body composition and in bone mineral density across decreasing BMI status. We suggest that changes in myostatin and in 25-hydroxyvitamin D levels may play a role in bone status of thin children. Significant correlations between adipokines and bone parameters may confirm crosstalk between fat tissue and bone.

## Figures and Tables

**Table 1 jcm-11-04013-t001:** Anthropometric parameters in the studied groups of children (group A—BMI z-score between +1 and −1 SD; group B—BMI z-score between −1 and −2 SD; group C—BMI z-score < −2 SD).

	Group A *n* = 35	Group B *n* = 36	Group C *n* = 10	*p*-Value
Age (years) ^a^	6.3 (5.0–8.0)	6.5 (5.6–7.5)	7.0 (5.4–8.7)	0.465
Weight (kg)	22.6 ± 5.8	19.5 ± 4.1	17.9 ± 4.2	0.004
Weight z-score	−0.05 ± 0.10	−1.09 ± 0.73	−1.94 ± 0.97	0.000
Height (cm)	119.9 ± 11.9	118.7 ± 10.6	118.9 ± 10.4	0.960
Height z-score	0.08 ± 0.23	−0.29 ± 0.28	−0.47 ± 0.44	0.401
BMI (kg/m^2^)	15.3 ± 1.0	13.6 ± 0.5	12.8 ± 0.7	0.000
BMI z-score	−0.14 ± 0.54	−1.40 ± 0.29	−2.61 ± 0.60	0.000
Fat mass (%)	22.2 ± 5.9	18.1 ± 5.6	16.1 ± 6.9	0.007
Fat mass (g)	4990 ± 1879	3150 ± 1180	2473 ± 726	0.000
Lean mass (g)	16,706 ± 5187	14,355 ± 3669	14,165 ± 4876	0.091
Fat/lean mass ratio	0.304 ± 0.097	0.229 ± 0.085	0.199 ± 0.101	0.003
tBMD (g/cm^2^)	0.819 ± 0.073	0.799 ± 0.106	0.694 ± 0.102	0.000
tBMD z-score	−0.010 ± 0.761	−0.278 ± 0.940	−0.780 ± 0.602	0.009
BMD L2-L4 (g/cm^2^)	0.698 ± 0.093	0.607 ± 0.099	0.558 ± 0.065	0.000
BMD L2-L4 z-score	−0.243 ± 0.684	−0.878 ± 0.982	−1.570 ± 0.695	0.001
tBMC (g)	732.5 ± 262.3	603.8 ± 187.8	550.6 ± 167,4	0.125

Data were analyzed using the non-parametric Kruskal–Wallis H test and presented as mean values ± SD or ^a^ medians and interquartile ranges (IQR); BMI—body mass index; tBMC—total bone mineral content; tBMD—total bone mineral density; BMD L2-L4—lumbar spine L2-L4 bone mineral density.

**Table 2 jcm-11-04013-t002:** Serum concentrations of myokines, adipokines and bone metabolism markers in the studied groups of children.

	Group A	Group B	Group C	*p*-Value
Resistin (ng/mL)	4.76 (3.28–5.67)	4.59 (3.71–5.41)	3.50 (2.34–4.49)	0.012
Visfatin (ng/mL)	1.41 (0.97–2.76)	1.07 (0.59–2.77)	0.91 (0.51–1.50)	0.228
Leptin (ng/mL)	1.77 (1.02–2.70)	1.24 (0.68–1.65)	1.03 (0.52–1.91)	0.128
Adiponectin (µg/mL)	8.78 (7.64–9.63)	7.95(5.77–10.52)	6.46(5.36–11.04)	0.287
HMW adiponectin (µg/mL)	5.46 (5.16–6.32)	4.93 (3.64–5.82)	3.43 (2.99–6.25)	0.020
HMW/adiponectin ratio	0.66 (0.53–0.71)	0.59 (0.49–0.65)	0.50 (0.41–0.62)	0.000
Omentin (ng/mL)	700 (298–1045)	976 (279–1269)	617 (279–976)	0.592
Myostatin (ng/mL)	0.72 (0.43–1.25)	0.83 (0.53- 1.25)	0.94 (0.52–1.44)	0.042
Irisin (µg/mL)	2.85 (2,30–4.15)	2.75 (2.27–4.80)	3.01 (2,13–4.58)	0.951
OC (ng/mL) ^a^	83.6 ± 0.5	77.4 ± 22.4	87.1 ± 23.9	0.355
CTX (ng/mL) ^a^	1.952 ± 0.430	1.935 ± 0.495	1.998 ± 0.504	0.913
OC/CTX ratio	0.45 (0.34–0.53)	0.42 (0.32–0.50)	0.44 (0.38–0.54)	0.647
25-hydroxyvitamin D (ng/mL) ^a^	29.1 ± 6.8	24.4 ± 7.5	24.1 ± 6.9	0.023

Data were analyzed using the non-parametric Kruskal–Wallis H test and presented as ^a^ mean values ± standard deviation (SD) or medians and interquartile ranges (IQR); HMW—high molecular weight; OC—osteocalcin; CTX—C-terminal telopeptide of collagen type I.

**Table 3 jcm-11-04013-t003:** The ratios of anti-inflammatory adipokines (adiponectin, omentin) to pro-inflammatory adipokines (leptin, resistin) in two studied groups of children (*n* = 71).

	Group A (*n* = 35)	Group B (*n* = 36)	*p*-Value
Adiponectin/leptin	6.94 (3.83–12.03)	5.90 (3.16–8.63)	0.168
Adiponectin/resistin	1.63 (1.19–2.31)	2.04 (1.60–2.70)	0.040
Omentin/leptin	5.73 (4.13–9.94)	4.11 (1.74–7.60)	0.045
Omentin/resistin	1.09 (0.54–2.75)	1.33 (0.66–2.58)	0.671

Data were analyzed using the Mann–Whitney U test and presented as medians and interquartile ranges (IQR).

**Table 4 jcm-11-04013-t004:** Correlations between body composition and bone mineral density with serum adipokines, myokines and bone metabolism markers in the whole group of children (*n* = 81).

	Fat Mass	Lean Mass	tBMC	tBMD	BMD L2-L4
BMI	0.660 ***	0.445 ***	0.425 ***	0.527 ***	0.452 ***
Fat mass	-	0.320 **	0.327 **	0.322 **	0.399 ***
Lean mass	-	-	0.878 ***	0.608 ***	0.518 ***
Leptin	0.262 *	0.398 ***	0.362 **	0.441 ***	0.317 **
Resistin	0.087	−0.251 *	0.166	−0.126	−0.078
Visfatin	0.050	−0.011	0.016	0.110	0.116
Adiponectin	−0.075	0.009	−0.061	0.034	0.005
HMW adiponectin	−0.205	0.006	−0.041	0.073	0.070
Omentin	−0.196	−0.101	0.009	0.061	0.073
Myostatin	0.028	−0.074	0.056	−0.160	−0.105
Irisin	−0.087	0.178	0.057	0.031	0.006
OC	0.219 *	0.185	0.188	0.135	0.133
CTX	−0.080	0.044	0.039	0.016	0.080
25-hydroxyvitamin D	0.305 **	0.036	0.080	0.081	0.152

Correlation analysis was performed using the Spearman (rho) coefficient; tBMC—bone mineral content; tBMD—bone mineral density in total body; BMD L2-L4—bone mineral density in lumbar spine L2-L4; BMI—body mass index; OC—osteocalcin; CTX—C-terminal telopeptide of collagen type I; * *p* < 0.05; ** *p* < 0.01; *** *p* < 0.001.

**Table 5 jcm-11-04013-t005:** Correlation analysis between ratios of anti- to pro-inflammatory adipokines and anthropometric parameters, bone markers and myokines in the whole group of studied children (*n* = 81).

	A/L Ratio		A/R Ratio		O/L Ratio		O/R Ratio	
	r	*p*	r	*p*	r	*p*	r	*p*
BMI	−0.218	0.046	0.121	0.284	−0.265	0.055	0.037	0.794
Fat mass	−0.059	0.599	−0.014	0.901	−0.268	0.054	−0.306	0.049
Lean mass	−0.370	0.001	0.147	0.191	−0.272	0.051	0.068	0.630
Fat/lean	0.198	0.077	−0.161	0.150	−0.082	0.566	−0.317	0.022
BMC	−0.371	0.001	0.068	0.551	−0.194	0.171	0.120	0.405
tBMD	−0.407	0.001	0.124	0.272	−0.366	0.008	0.133	0.349
BMD L2-L4	−0.318	0.004	0.064	0.570	−0.270	0.053	0.091	0.523
OC	−0.140	0.211	0.160	0.153	−0.034	0.811	0.097	0.494
CTX	0.048	0.668	0.052	0.645	−0.046	0.745	0.032	0.819
25-hydroxyvitamin D	0.068	0.548	0.039	0.728	−0.233	0.097	−0.304	0.048
Myostatin	−0.088	0.437	0.128	0.256	0.216	0.123	0.326	0.019
Irisin	−0.020	0.860	0.161	0.151	0.214	0.128	0.292	0.036

Correlation analysis was performed using the Spearman (rho) coefficient; A/L—total adiponectin/leptin; A/R—total adiponectin/resistin; O/L—omentin/leptin; O/R—omentin/resistin.

## Data Availability

All the data have been provided in the manuscript.

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
