# Peer review of "Relationships between Body Weight Status and Serum Levels of Adipokine, Myokine and Bone Metabolism Parameters in Healthy Normal Weight and Thin Children"

_jcm, 2022, doi:10.3390/jcm11144013_

Round 1

Reviewer 1 Report

The authors present a well-written manuscript in which they highlight the relationships between body weight and adipokine, myokine and bone metabolism among a “normal” group of pubertal children, in which they state a good base to the understanding of the muscle-bone-fat relation.

In order to improve the method and discussion sections, the authors could describe the routine vitamin D supplementation, (dose and frequency), as it could be different or not usual in similar populations. In this sense, it would be helpful for readers to have a little discussion regarding the found differences in concentrations of 25-hydroxyvitamin D among groups (A vs B&C), and if they are of clinical importance. In this sense, it could be useful to readers if the authors try to explain the mechanisms in which it has been reported a greater bone mass among children with obesity despite these factors, as a trend was seen with their data.

Also, regarding physical activity how the authors assure they were attaining the WHO recommednations, as it has been suggested and increase in sedentary time across the world. As this two variables influence directly over the bone health, weight status and therefore the synthesis of myokines.

Despite the design of the study, the proposed analysis are approppiate and suggest a need to further research on this area, just to be more clear about the methology,  It could be useful for readers to know more about  the sampling or the enrollment to this analysis, trying to explain and discuss if the results were diminished for the size sample or the fact that no obese or overweight children were included, albeit is a population were suppositions of this relationships have been suggested (even in the proposed title of the manuscript where it is not stated that is a "healthy" group without all weight variants ).

Author Response

The authors present a well-written manuscript in which they highlight the relationships between body weight and adipokine, myokine and bone metabolism among a “normal” group of pubertal children, in which they state a good base to the understanding of the muscle-bone-fat relation.

Thank you very much for your valuable comments and suggestions which substantially improved the quality of this manuscript. We have taken into account all your comments.

In order to improve the method and discussion sections, the authors could describe the routine vitamin D supplementation, (dose and frequency), as it could be different or not usual in similar populations. In this sense, it would be helpful for readers to have a little discussion regarding the found differences in concentrations of 25-hydroxyvitamin D among groups (A vs B&C), and if they are of clinical importance. In this sense, it could be useful to readers if the authors try to explain the mechanisms in which it has been reported a greater bone mass among children with obesity despite these factors, as a trend was seen with their data.

In our country vitamin D insufficiency in both adults and children is particularly common. Therefore, the routine supplementation of vitamin D in children is recommended, especially in the autumn and winter period [Rusińska A et al. Vitamin D supplementation guidelines for Poland – A 2018 updata. Postępy Neonatologii 2018, 24(1)].

            The studied children were supplemented (except of 3 cases) with vitamin D at a median dose of 1000 (range 600-1500 IU/day). Despite the supplementation, mean serum concentration of 25-hydroxyvitamin D in children did not reach an optimal levels of this vitamin (30-50 ng/ml), especially in the group of children with lower body weight status (about 24 ng/ml in group B and C). This may have an impact on bone metabolism in these children.

We added information about supplementation with vitamin D in examined children and little discussion regarding vitamin D and bone mass in obese children. In the present study, observed changes in vitamin D and myostatin levels may be associated with lower bone mineral density in thin children. This is the first study that investigated adipokines, myokines and their impact on bone status in prepubertal children across decreasing body weight status, therefore further examination of larger groups of children is needed to identify these associations.

We included this information in the Materials and Methods and Discussion sections.

Also, regarding physical activity how the authors assure they were attaining the WHO recommednations, as it has been suggested and increase in sedentary time across the world. As this two variables influence directly over the bone health, weight status and therefore the synthesis of myokines.

It is known that physical activity is associated with health benefits and has a very important effect on bone status as well as on myokines synthesis. According to the WHO guidelines regarding physical activity [WHO guidelines on physical activity and sedentary behaviour. Geneva: World Health Organization; 2020. Licence: CC BY-NC-SA 3.0 IGO], children and adolescents aged 5-17 years should accumulate 60 minutes of moderate-to-vigorous physical activity (MVPA) each day. Evidence also suggests that they should engage in vigorous physical activity (VPA) at least 3 days a week.

We have only limited information about physical activity, because it was not the main aim of the present study. However, we recognized (by a questionnaire) that the level of physical activity was similar in the studied groups. The examined children were attaining these recommendations and accumulated about 60-90 minutes daily of MVPA and approximately 30 minutes of VPA (it was activities after school twice a week for 1 or 2 hours).

We mentioned the similar level of physical activity in the studied groups of children in the Materials and Methods and Discussion sections.

Despite the design of the study, the proposed analysis are approppiate and suggest a need to further research on this area, just to be more clear about the methology,  It could be useful for readers to know more about  the sampling or the enrollment to this analysis, trying to explain and discuss if the results were diminished for the size sample or the fact that no obese or overweight children were included, albeit is a population were suppositions of this relationships have been suggested (even in the proposed title of the manuscript where it is not stated that is a "healthy" group without all weight variants ).

We agree with Reviewer`s suggestions that further research on this area in larger groups of subjects is needed. We realize that large groups of healthy children are rarely studied because of ethical considerations regarding blood sampling in healthy children. In our study, we examined only groups of normal weight and thin children, however, we are planning to investigated overweight and obese children in term of adipokine and myokine levels and their relationships with bone parameters.

We changed the title and aim of this study to narrow down the research question.

English language was checked.

Reviewer 2 Report

The authors examined the relationships between body weight status and serum levels of adipokine, myokine and bone metabolism parameters among healthy prepubertal children. While this study provides important evidence, there are some concerns that need to be addressed. 

Major comments

1.       Although many associations have been examined and results presented in this study, the research question could be narrowed down.

2.       Abstract Line 33 Lines 326, 393

It is unclear how this statement was derived. This is a cross-sectional study, and the authors should be cautious about referring to mechanisms.

3.       Line 297

It is unclear how this statement was derived. This is a cross-sectional study, and the authors should be cautious about referring to mechanisms.

4.       Line 338

This sentence needs an explanation. Please add an explanation.

5.       Line 179

The authors should do a non-parametric correlation analysis instead of the Pearson method.

6.       A statement should be added in the text whether there is any diurnal or seasonal variation in each adipokine and myokine.

7.       In the discussion, adipokines are divided into anti-inflammatory (Adiponectin, Omentin ) and pro-inflammatory (leptin and resistin), and their position should be clearly stated in the introduction.

Minor comments

1.       Line 121 World Health Organization's is unnecessary, only WHO is acceptable. 

Line 338 typo z-score.

2.       The subscripts in the table should be superscripted. E.g. Age (year) a, kg/m2

The notation of Vitamin D in the table should be unified. 20-OH D, 25-hydroxyvitamin D, Vitamin D

Author Response

The authors examined the relationships between body weight status and serum levels of adipokine, myokine and bone metabolism parameters among healthy prepubertal children. While this study provides important evidence, there are some concerns that need to be addressed. 

Thank you very much for your valuable comments and suggestions which substantially improved the quality of this manuscript. We corrected all issues point by point.

Major comments

  1. Although many associations have been examined and results presented in this study, the research question could be narrowed down.

 We changed the title and aim of this study (in Abstract and in Introduction) to narrow down the research question.

  1. 2.Abstract Line 33 Lines 326, 393

It is unclear how this statement was derived. This is a cross-sectional study, and the authors should be cautious about referring to mechanisms.

We agree with the Reviewer`s suggestion and changed the statement in the lines 31-35, and 447-452.

  1. 3.Line 297

It is unclear how this statement was derived. This is a cross-sectional study, and the authors should be cautious about referring to mechanisms.

According to the Reviewer` suggestion, we changed this statement (line 334-335).

  1. Line 338

This sentence needs an explanation. Please add an explanation.

We added an explanation in lines 338-392.

  1. 5.Line 179

The authors should do a non-parametric correlation analysis instead of the Pearson method.

We performed correlation analysis using the non-parametric Spearman test instead of the Pearson test (changed data in the Table 4, Table 5 and in the text).

  1. 6.A statement should be added in the text whether there is any diurnal or seasonal variation in each adipokine and myokine.

      The stability of circulating adipokine and myokine levels within individuals, and their seasonal variability, are not fully recognized. The researchers reported negative association between adiponectin levels and winter-summer difference in self-reported sleep duration and between adiponectin levels and self-reported seasonal change in weight [Akram et al. Seasonal affective disorder and seasonal changes in weight and sleep duration are inversely associated with plasma adiponectin levels. J Psychiatr Res. 2020, 122:97-104. doi: 10.1016/j.jpsychires.2019.12.016].

In healthy young humans leptin has a high intra-individual stability and seasonal leptin variation does not appear [Anastasilakis et al. Circulating follistatin displays a day-night rhythm and is associated with muscle mass and circulating leptin levels in healthy, young humans. Metabolism, 2016, 65(10):1459-65. doi: 10.1016/j.metabol.2016.07.002].

From animal studies (brown bears) it is known that adipokine concentrations were relatively low throughout the active season but peaked in mid-October prior to hibernation when fat content was greatest [Rigano et al. Life in the fat lane: seasonal regulation of insulin sensitivity, food intake, and adipose biology in brown bears.  J Comp Physiol B. 2017, 187(4):649-676. doi: 10.1007/s00360-016-1050-9].

To avoid diurnal variation of biochemical parameters, blood samples were collected in the morning hours between 8:00 and 10:00 AM after an overnight fast.

This statement is included in the Materials and Methods section.

  1. In the discussion, adipokines are divided into anti-inflammatory (Adiponectin, Omentin ) and pro-inflammatory (leptin and resistin), and their position should be clearly stated in the introduction.

We added information about pro- and anti-inflammatory properties of adipokines in Introduction and re-written a part of this section.

Minor comments

  1. Line 121 “World Health Organization's” is unnecessary, only “WHO” is acceptable.

Line 338 typo z-score.

  1. The subscripts in the table should be superscripted. g. Age (year) a, kg/m2

The notation of Vitamin D in the table should be unified. 20-OH D, 25-hydroxyvitamin D, Vitamin D

We corrected all suggested minor points in the manuscript (in the text and in tables). We unified notation of vitamin D as 25-hydroxyvitamin D.

The authors examined the relationships between body weight status and serum levels of adipokine, myokine and bone metabolism parameters among healthy prepubertal children. While this study provides important evidence, there are some concerns that need to be addressed. 

Thank you very much for your valuable comments and suggestions which substantially improved the quality of this manuscript. We corrected all issues point by point.

Major comments

  1. Although many associations have been examined and results presented in this study, the research question could be narrowed down.

 We changed the title and aim of this study (in Abstract and in Introduction) to narrow down the research question.

  1. 2.Abstract Line 33 Lines 326, 393

It is unclear how this statement was derived. This is a cross-sectional study, and the authors should be cautious about referring to mechanisms.

We agree with the Reviewer`s suggestion and changed the statement in the lines 31-35, and 447-452.

  1. 3.Line 297

It is unclear how this statement was derived. This is a cross-sectional study, and the authors should be cautious about referring to mechanisms.

According to the Reviewer` suggestion, we changed this statement (line 334-335).

  1. Line 338

This sentence needs an explanation. Please add an explanation.

We added an explanation in lines 338-392.

  1. 5.Line 179

The authors should do a non-parametric correlation analysis instead of the Pearson method.

We performed correlation analysis using the non-parametric Spearman test instead of the Pearson test (changed data in the Table 4, Table 5 and in the text).

  1. 6.A statement should be added in the text whether there is any diurnal or seasonal variation in each adipokine and myokine.

      The stability of circulating adipokine and myokine levels within individuals, and their seasonal variability, are not fully recognized. The researchers reported negative association between adiponectin levels and winter-summer difference in self-reported sleep duration and between adiponectin levels and self-reported seasonal change in weight [Akram et al. Seasonal affective disorder and seasonal changes in weight and sleep duration are inversely associated with plasma adiponectin levels. J Psychiatr Res. 2020, 122:97-104. doi: 10.1016/j.jpsychires.2019.12.016].

In healthy young humans leptin has a high intra-individual stability and seasonal leptin variation does not appear [Anastasilakis et al. Circulating follistatin displays a day-night rhythm and is associated with muscle mass and circulating leptin levels in healthy, young humans. Metabolism, 2016, 65(10):1459-65. doi: 10.1016/j.metabol.2016.07.002].

From animal studies (brown bears) it is known that adipokine concentrations were relatively low throughout the active season but peaked in mid-October prior to hibernation when fat content was greatest [Rigano et al. Life in the fat lane: seasonal regulation of insulin sensitivity, food intake, and adipose biology in brown bears.  J Comp Physiol B. 2017, 187(4):649-676. doi: 10.1007/s00360-016-1050-9].

To avoid diurnal variation of biochemical parameters, blood samples were collected in the morning hours between 8:00 and 10:00 AM after an overnight fast.

This statement is included in the Materials and Methods section.

  1. In the discussion, adipokines are divided into anti-inflammatory (Adiponectin, Omentin ) and pro-inflammatory (leptin and resistin), and their position should be clearly stated in the introduction.

We added information about pro- and anti-inflammatory properties of adipokines in Introduction and re-written a part of this section.

Minor comments

  1. Line 121 “World Health Organization's” is unnecessary, only “WHO” is acceptable.

Line 338 typo z-score.

  1. The subscripts in the table should be superscripted. g. Age (year) a, kg/m2

The notation of Vitamin D in the table should be unified. 20-OH D, 25-hydroxyvitamin D, Vitamin D

We corrected all suggested minor points in the manuscript (in the text and in tables). We unified notation of vitamin D as 25-hydroxyvitamin D.

Round 2

Reviewer 2 Report

The authors fully addressed the reviewers' points.